# Understanding Catastrophic Interference On the Identifibility of Latent Representations

## Abstract

Catastrophic interference, also known as catastrophic forgetting, is a fundamental challenge in machine learning, where a trained learning model progressively loses performance on previously learned tasks when adapting to new ones. In this paper, we aim to better understand and model the catastrophic interference problem from a latent representation learning point of view, and propose a novel theoretical framework that formulates catastrophic interference as an identification problem. Our analysis demonstrates that the forgetting phenomenon can be quantified by the distance between partial-task aware (PTA) and all-task aware (ATA) setups. Building upon recent advances in identifiability theory, we prove that this distance can be minimized through identification of shared latent variables between these setups. When learning, we propose our method **ICON** with two-stage training strategy: First, we employ maximum likelihood estimation to learn the latent representations from both PTA and ATA configurations. Subsequently, we optimize the KL divergence to identify and learn the shared latent variables. Through theoretical guarantee and empirical validations, we establish that identifying and learning these shared representations can effectively mitigate catastrophic interference in machine learning systems. Our approach provides both theoretical guarantees and practical performance improvements across both synthetic and benchmark datasets.

## 1 Introduction

catastrophic interference represents a fundamental challenge in machine learning (Cha et al., 2021; Liang & Li, 2023; Xiao et al., 2024), where a model trained sequentially on multiple tasks experiences significant performance degradation on previously learned tasks when adapting to new ones. This phenomenon manifests as a direct consequence of the distributional shift between tasks, coupled with the model's capability to preserve previously learned knowledge when optimizing for new data. Therefore, the model must adapt to new tasks while preserving critical knowledge from earlier experiences, mirroring human cognitive abilities to accumulate knowledge progressively.

Handling catastrophic interference presents unique theoretical challenges, as the model involves a dynamic evolution of itself as it learns new tasks. This creates an inherent instability in the data representation process —learning from new tasks fundamentally alters the model's parameters, potentially disrupting the representations learned for previous tasks. In other words, the learned data generating process changes from task to task as the models evolve. To better understand and model this challenge, we take inspiration from recent advances in Causal Representation Learning (Schölkopf et al., 2021; Kong et al., 2022; Li et al., 2023; Kong et al., 2024), and approach this problem via modeling the data generating process through the mixing functions that map low-dimensional latent variables to high-dimensional observations. More specifically, we distinguish between two configurations in catastrophic interference handling: the partial-task aware (PTA) setting, which represents a model trained on a subset of tasks, uses a mixing function that has only seen data up to the current task, and the all-task aware (ATA) setting, which represents an ideal model trained on all tasks, leverages a single task-invariant mixing function $g$.

In this work, we propose a new theoretical framework that formulates catastrophic interference as a latent-variable identification problem. Our key insight is that catastrophic interference can be quantified by measuring the distance between latent representations in PTA and ATA settings. By

identifying the shared latent variables between these setups, we can establish a principled approach to preserving knowledge across distributional shifts. Builds upon our theoretical findings, we introduce an a two-stage learning methodology, Identifiable CatastrOphic iNterference (`ICON`). First, we employ maximum likelihood estimation to learn the latent representations from both PTA and ATA configurations independently. Subsequently, we optimize the KL divergence between these representations to identify and learn their shared components. Through evaluating on both synthetic data and real-world benchmarks, `ICON` effectively mitigate the catastrophic interference.

Our contributions are threefold: (1) We formulate catastrophic interference as a latent-variable identification problem, providing a novel theoretical perspective that quantifies forgetting through distributional distances; (2) We establish identifiability conditions for the shared latent variables between PTA and ATA setups, proving when and how knowledge can be preserved across distributional shifts under certain assumptions; (3) Based on our theoretical findings, we develop a practical approach that demonstrates superior performance on both synthetic data and standard benchmarks on handling catastrophic interference, outperforming current state-of-the-art methods. By bridging theory and practice, our work provides the first work delving into the nature of catastrophic interference through identifications, and offers a principled framework.

## 2 RELATED WORK

### 2.1 HANDLING CATASTROPHIC FORGETTING

Existing learning methods can be categorized into five primary approaches when handling catastrophic interference: (1) *Regularization-based methods* introduce constraints on model parameters or outputs within the loss function to mitigate catastrophic forgetting when learning new tasks. Representative works include Chaudhry et al. (2018); Aljundi et al. (2018); Hou et al. (2019); Cha et al. (2021). (2) *Memory replay-based methods* explicitly store and revisit past experiences by maintaining a subset of previous task samples, thereby reducing forgetting. Notable examples include Arani et al. (2022); Caccia et al. (2022); Bonicelli et al. (2022); Sarfraz et al. (2023); Wang et al. (2023b); Liang & Li (2023). (3) *Gradient-projection-based methods* mitigate forgetting by constraining gradient updates to subspaces that minimize interference with prior knowledge. Relevant studies include Chaudhry et al. (2020); Farajtabar et al. (2020); Saha et al. (2021); Wang et al. (2021); Lin et al. (2022); Qiao et al. (2024); Xiao et al. (2024). (4) *Architecture-based methods* dynamically adjust the neural network structure to integrate new tasks while preserving performance on previous ones. Key contributions in this category include Mallya & Lazebnik (2018); Serra et al. (2018); Li et al. (2019); Hung et al. (2019). (5) *Bayesian-based methods* leverage Bayesian inference principles to model uncertainty and facilitate new task while maintaining prior knowledge. Representative works include Kao et al. (2021); Henning et al. (2021); Pan et al. (2020); Titsias et al. (2020); Rudner et al. (2022).

### 2.2 IDENTIFIABILITY OF LATENT VARIABLES WITH DISTRIBUTION SHIFTS

Identifying latent variables in causal representation learning has emerged as a foundational paradigm for understanding representation learning in deep neural networks (Schölkopf et al., 2021; Khemakhem et al., 2020). This approach typically assumes latent variables $\mathbf{z}$ generate observed data $\mathbf{x}$ through a generative function. However, when this function exhibits nonlinearity—as is common in deep learning models—recovering the original latent variables becomes technically challenging (Khemakhem et al., 2020).

To address this challenge, several recent works (Li et al., 2023; Song et al., 2023; Chen et al., 2024; Zheng & Zhang, 2024; Morioka & Hyvarinen, 2024) have introduced auxiliary labels $\mathbf{u}$ that induce distributional shifts in the latent components across different conditions. While effective in certain scenarios, these approaches depend critically on assuming the access to multiple disparate distributions with overlapping supports, including the target distribution. Furthermore, recent advances by (Yao et al., 2024; Kong et al., 2024) either require labeled grouping data generating process or assume identical mixing functions across different data generating processes. Our theoretical framework overcomes these limitations by eliminating the need for either labeled diverse generating processes or identical mixing functions. This broadened scope encompasses earlier work such as (Yao et al., 2024; Kong et al., 2024) as a special case or ours.

# 3 PROBLEM SETUP

Given $T$ tasks in total, we aim to learn a task-invariant model to adapt to all tasks. However, the possible data distributions shift across tasks raises the challenge of catastrophic forgetting, where a model's performance on previously learned task $\mathbf{t}$ could degrade after training on all $T$ tasks.

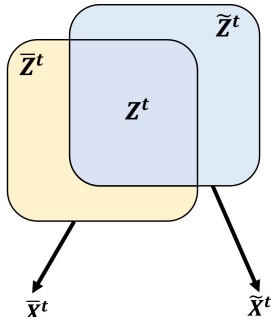

In this section, to formally characterize the nature of catastrophic forgetting, we present two data generating processes.

We term the first one by partial-task aware (PTA) approach, leveraging the mixing functions $g^{:\mathbf{t}}$ from task 1 to task $t-1$ ($t > 1$), resulting $T-1$ mixing functions of PTA setup in total:

$$\mathbf{x}^{\mathbf{t}} = g^{:\mathbf{t}}(\overline{\mathbf{z}}^{\mathbf{t}}) \tag{1}$$

where $\mathbf{x}^{\mathbf{t}} \in \mathbb{R}^K$ denotes the observations of task $\mathbf{t}$, the nonlinear mixing function $g^{:\mathbf{t}} : \mathbb{R}^N$ is a diffeomorphism onto $\mathbb{R}^K$, and $\overline{\mathbf{z}}^{\mathbf{t}} \in \mathbb{R}^N$ denotes the task-specific continuous latent variable.

Unlike the PTA approach, the second data generating process represents the all-task aware (ATA) paradigm, which aims to learn a mixing function $g$ that can handle all $T$ tasks. ATA meets the goal of continual learning in the sense that it works on all domains:

$$\mathbf{x}^{\mathbf{t}} = g(\tilde{\mathbf{z}}^{\mathbf{t}}) \tag{2}$$

Figure 1: The illustration of our definition of subspace identification in Def. 3. Given two space of latent variables of PTA setup $\overline{\mathcal{Z}}^{\mathbf{t}} \ni \overline{\mathbf{z}}^{\mathbf{t}}$ and ATA approach $\tilde{\mathcal{Z}}^{\mathbf{t}} \ni \tilde{\mathbf{z}}^{\mathbf{t}}$, we aim to identify their intersection $\mathbf{z}^{\mathbf{t}} \in \mathcal{Z}^{\mathbf{t}} = \overline{\mathcal{Z}}^{\mathbf{t}} \cap \tilde{\mathcal{Z}}^{\mathbf{t}}$.

Simialrly, $\tilde{\mathbf{z}}^{\mathbf{t}} \in \mathbb{R}^N$ denotes the continous latent variable for the task $\mathbf{t}$, $g$ is an nonlinear mixing function and diffeomorphism onto $\mathbb{R}^K$. For both Eqs. 1 and 2, we focus on the undercomplete case, i.e., $N \leq K$.

We are now ready to connect catastrophic forgetting with Eq. 1 and 2. Let us define a $\mathcal{G} \subset g^{:t} : \mathbb{R}^{\overline{\mathbf{z}}} \to \mathbb{R}^{\mathbf{x}}$, where $G$ denotes a hypothesis class. $l : \mathcal{G} \times \mathbb{R}^{\mathbf{x}} \to [0, B_l]$ denotes the loss function, where $B_l > 0$ is a constant. In this work, we leverage the negative log-likelihood for $l$, i.e., $l(\hat{g}^{:t}, \mathbf{x}^{\mathbf{t}}) = -\log p_{\hat{g}^{:t}}(\mathbf{x}^{\mathbf{t}})$. Similar, we define a lost function $l(\hat{g}, \mathbf{x}^{\mathbf{t}}) = -\log p_{\hat{g}}(\mathbf{x}^{\mathbf{t}})$ Following the defintion 1.1 in Wang et al. (2024), we reinterpret the catastrophfic forgetting $\mathcal{F}$ by:

$$\mathcal{F} = \mathbb{E}_{\mathbf{t}} \frac{1}{|\mathbf{x}^{\mathbf{t}}|} \sum_{\mathbf{x}^{\mathbf{t}}} (-l(\hat{g}^{:t}, \mathbf{x}^{\mathbf{t}}) + l(g, \mathbf{x}^{\mathbf{t}})) \tag{3}$$

where $|\mathbf{x}^{\mathbf{t}}|$ denotes the sample numbers of $\mathbf{x}^{\mathbf{t}}$. This formulation aligns with Definition 1.1 in Wang et al. (2024) by utilizing negative log-likelihood as the performance measurement metric, i.e., Eq. 3 quantifies the difference between two data generating process from Eq. 1 and Eq. 2, respectively.

Given the invertibility of both mapping functions $g^{:\mathbf{t}}$ and $g$, and both $\overline{\mathbf{z}}^{\mathbf{t}}$ and $\tilde{\mathbf{z}}^{\mathbf{t}}$ live in $\mathbb{R}^N$, we establish the observed differences under Eq. 2 and Eq. 1 uniquely determined by the underlying differences between their respective latent representations $\overline{\mathbf{z}}^{\mathbf{t}}$ and $\tilde{\mathbf{z}}^{\mathbf{t}}$. This allows us to decompose $\overline{\mathbf{z}}^{\mathbf{t}}$ and $\tilde{\mathbf{z}}^{\mathbf{t}}$ into two parts: their difference and their overlap. In this work, to minimize $\mathcal{F}$ in Eq. 3, we focus on identifying the overlap between $\overline{\mathbf{z}}^{\mathbf{t}}$ and $\tilde{\mathbf{z}}^{\mathbf{t}}$ across the PTA and ATA settings. Formally, for the latent variable manifolds $\overline{\mathcal{Z}}^{\mathbf{t}} \ni \overline{\mathbf{z}}^{\mathbf{t}}$ and $\tilde{\mathcal{Z}}^{\mathbf{t}} \ni \tilde{\mathbf{z}}^{\mathbf{t}}$, we denote their intersection by $\mathcal{Z}^{\mathbf{t}} = \overline{\mathcal{Z}}^{\mathbf{t}} \cap \tilde{\mathcal{Z}}^{\mathbf{t}}$. $\forall \mathbf{z}^{\mathbf{t}} \in \mathcal{Z}^{\mathbf{t}}, \mathbf{z}^{\mathbf{t}} = \tilde{\mathbf{z}}^{\mathbf{t}} = \overline{\mathbf{z}}^{\mathbf{t}}$. We introduce the definition of the identifiability in the following (subspace identifiability suffices):

(**Identifiability of latent variables shared by Eqs. 1 and 2**): *For any pair of mixing functions $(g^t, g)$ defined in Equations 1 and 2 respectively, there exists a shared latent space region $\mathcal{Z}^{\mathbf{t}}$ where the latent variables from both the PTA and ATA settings coincide. $\forall \mathbf{z}^{\mathbf{t}} \in \mathcal{Z}^{\mathbf{t}}$, there exists an invertible transformation $t$ such that: $\hat{\mathbf{z}}^{\mathbf{t}} = t(\mathbf{z}^{\mathbf{t}})$.*

# 4 IDENTIFIABILITY THEORY

To establish our identifiability results, we begin with formalizing the distance between two manifolds $\mathcal{Z}_1^{\mathbf{t}}$ and $\mathcal{Z}_2\mathbf{t}$ as:

$$\mathcal{D}(\mathcal{Z}_1^{\mathbf{t}}, \mathcal{Z}_2^{\mathbf{t}}) = inf_{\mathbf{z}'_1 \in \mathcal{Z}_1^{\mathbf{t}}, \mathbf{z}'_2 \in \mathcal{Z}_2^{\mathbf{t}}} \|\mathbf{z}'_1 - \mathbf{z}'_2\|_2 \tag{4}$$

where $\overline{\mathbf{z}}'$ denotes points on the boundary support of $\overline{\mathcal{Z}}^{\mathbf{t}}$, and $\tilde{\mathbf{z}}'$ denotes points on the boundary support of $\tilde{\mathcal{Z}}^{\mathbf{t}}$. Here, $\| \cdot \|_2$ represents the Euclidean distance.

We define the minimum distance from a point $\mathbf{z}^{\mathbf{t}}$ to any point in the manifold $\tilde{\mathcal{Z}}^{\mathbf{t}}$ for the ATA setting:

$$\mathcal{D}(\tilde{\mathbf{z}}^{\mathbf{t}}, \mathbf{z}^{\mathbf{t}}) = \arg \min_{\tilde{\mathbf{z}}^{\mathbf{t}} \in \tilde{\mathcal{Z}}^{\mathbf{t}}} \|\tilde{\mathbf{z}}^{\mathbf{t}} - \mathbf{z}^{\mathbf{t}}\|_2 \tag{5}$$

Similarly, for the PTA setting, we define:

$$\mathcal{D}(\overline{\mathbf{z}}^{\mathbf{t}}, \mathbf{z}^{\mathbf{t}}) = \arg \min_{\overline{\mathbf{z}}^{\mathbf{t}} \in \overline{\mathcal{Z}}^{\mathbf{t}}} \|\overline{\mathbf{z}}^{\mathbf{t}} - \mathbf{z}^{\mathbf{t}}\|_2 \tag{6}$$

Having established the necessary distance metrics, we are now ready to present our identifiability results.

**Theorem 1.** *Given the data generating process in Eq. 1 and Eq. 2, if the following assumptions are satisfied:*

1. *(Smoothness and invertibility) The mixing function $g^{\mathbf{t}}$ in Eq. 1, and $g$ in Eq. 2 are smooth functions and invertible everywhere;*

2. *(The existence of intersection) For the latent variable manifolds $\overline{\mathcal{Z}}^{\mathbf{t}} \ni \overline{\mathbf{z}}^{\mathbf{t}}$ and $\tilde{\mathcal{Z}}^{\mathbf{t}} \ni \tilde{\mathbf{z}}^{\mathbf{t}}$, their intersection $\mathcal{Z}^{\mathbf{t}} = \overline{\mathcal{Z}}^{\mathbf{t}} \cap \tilde{\mathcal{Z}}^{\mathbf{t}} \neq \emptyset$.*

3. *(Path-connected) both $\overline{\mathcal{Z}}^{\mathbf{t}}$ and $\tilde{\mathcal{Z}}^{\mathbf{t}}$ are path-connected;*

4. *(Compactness) The spaces of observed variables $\overline{\mathbf{x}}^{\mathbf{t}}$ and $\tilde{\mathbf{x}}^{\mathbf{t}}$ are closed and bounded;*

5. *(Constrained out-of-intersection distance) $\forall \tilde{\mathbf{z}}^{\mathbf{t}} \notin \mathcal{Z}^{\mathbf{t}}$, if $\exists \tilde{\mathcal{Z}}_1^{\mathbf{t}}, \tilde{\mathcal{Z}}_2^{\mathbf{t}}$, the distance $\mathcal{D}(\tilde{\mathbf{z}}^{\mathbf{t}}, \mathbf{z}^{\mathbf{t}})$ between the out-of-intersection $\tilde{\mathbf{z}}^{\mathbf{t}}$ and $\mathbf{z}^{\mathbf{t}}$ is constrained by $\mathcal{D}(\tilde{\mathbf{z}}^{\mathbf{t}}, \mathbf{z}^{\mathbf{t}}) \leq \frac{\mathcal{D}(\tilde{\mathcal{Z}}_1^{\mathbf{t}}, \tilde{\mathcal{Z}}_2^{\mathbf{t}})}{2 J_{\tilde{\mathbf{u}}}}$. $J_{\tilde{\mathbf{u}}}$ denotes the spectrum norm of $\{J_g(\tilde{\mathbf{z}}_1^{\mathbf{t}}), J_g(\tilde{\mathbf{z}}_2^{\mathbf{t}})\}$.*

*If we can learn the optimal estimation $\hat{\mathbf{z}}^{\mathbf{t}}$ of $\mathbf{z}^{\mathbf{t}} \in \mathcal{Z}^{\mathbf{t}}$ such that:*

$$sup(p(\hat{\mathbf{z}})), \ s.t. \ sup(p_{\hat{g}^t}(\mathbf{x}^{\mathbf{t}})) \quad \& \quad sup(p_{\hat{g}}(\mathbf{x}^{\mathbf{t}})) \tag{7}$$

*then the identifiability results stated in Def. 3 are obtained.*

**Proof Sketch:** Our proof establishes identifiability via contradiction by assuming a latent variable $\mathbf{z}^t$ simultaneously resides on two distinct latent manifolds. Differences between latent points are expressed as integrals involving the Jacobian of the generative function $g$, bounded by the spectral norm of the Jacobian multiplied by their latent-space distance. Since both latent points generate the same observed data, their output difference must be zero, implying a zero distance between distinct latent points. This result contradicts the theorem's assumption of a strictly positive minimal separation between distinct manifold points. Therefore, each observed data point must originate from a unique latent point, ensuring identifiability.

**Remarks:** Assumption 1 & Assumption 2 guarantee the the existence of an intersection between $\overline{\mathcal{Z}}$ and $\tilde{\mathcal{Z}}$. The path-connectedness specified in Assumption 3 and The compactness property in Assumption 4 impose the geometric boundedness of the overlap between $\overline{\mathcal{Z}}$ and $\tilde{\mathcal{Z}}$. Assumption 5 establishes critical upper bounds for the distances $\mathcal{D}(\tilde{\mathbf{z}}^{\mathbf{t}}, \mathbf{z}^{\mathbf{t}})$.

## 5 **ICON** APPROACH

Building upon our identifiability results, we now introduce **ICON** to estimate the latent causal variables. Our approach aims to achieve the observational equivalence in Eq. 7 by modeling the data generating processes in Eqs. 1 and 2. In what follows, we introduce each part of our network individually.

### 5.1 NETWORK DESIGN

Our network architecture is designed to uncover the latent variables for both PTA and ATA setup through carefully constructed flow-based models.

**Network Structure for PTA approach**   For the PTA setup described in Eq. 1, `ICON` formalizes the probabilistic joint distribution as:

$$p(\mathbf{x^t}, \overline{\mathbf{z}}^\mathbf{t}) = p_{g^{\cdot\mathbf{t}}}(\mathbf{x^t}|\overline{\mathbf{z}}^\mathbf{t})p(\overline{\mathbf{z}}^\mathbf{t}) \tag{8}$$

To implement the invertible mapping function $g^\mathbf{t}$ in Eq. 1, we employ General Incompressible-flow Network (GIN) Sorrenson et al. (2020), which provide a highly expressive class of normalizing flows with the following inverse mapping:

$$\hat{\overline{\mathbf{z}}}^\mathbf{t} \sim \mathcal{N}(\hat{\overline{\mu}}^\mathbf{t}, \hat{\overline{\sigma}}^\mathbf{t}), \; \hat{\overline{\mu}}^\mathbf{t}, \hat{\overline{\sigma}}^\mathbf{t} = \hat{g}_{-1}^{:t}(\mathbf{x^t}) \tag{9}$$

where $\hat{g}_{-1}^{:\mathbf{t}}$ denotes the estimation of inverse of the mixing function $g^\mathbf{t}$ for PTA settings.

**Network Structure for ATA framework**   In contrast to the PTA setting, the ATA framework in Eq. 2 requires to learn a task-invariant mixing function $g$, The joint distribution in this case is modeled as:

$$p(\mathbf{x^t}, \tilde{\mathbf{z}}^\mathbf{t}) = p_g(\mathbf{x^t}|\tilde{\mathbf{z}}^\mathbf{t})p(\tilde{\mathbf{z}}^\mathbf{t})d \tag{10}$$

We implement this task-invariant mapping using another GIN model that processes data from all tasks:

$$\hat{\tilde{\mathbf{z}}}^\mathbf{t} \sim \mathcal{N}(\hat{\tilde{\mu}}^\mathbf{t}, \hat{\tilde{\sigma}}^\mathbf{t}), \; \hat{\tilde{\mu}}^\mathbf{t}, \hat{\tilde{\sigma}}^\mathbf{t} = \hat{g}_{-1}(\mathbf{x^t}) \tag{11}$$

## 5.2 Learning Objective

In this work, we learn $\hat{\overline{\mathbf{z}}}^\mathbf{t}$ through maximum likelihood estimation (MLE). Specifically, we estimate $p(\hat{\mathbf{x}}^\mathbf{t})$ of Eq. 8 by optimizing the following objective:

$$\mathcal{L}^\mathbf{t} = \frac{1}{\mathbf{t}} \sum_{i=1}^{\mathbf{t}} \Big( \frac{1}{|\mathbf{x^t}|} \sum_{\mathbf{x^t}} \big( \log p(\hat{g}_{-1}^{:\mathbf{t}}(\mathbf{x^t})) \big) \Big) \tag{12}$$

where $|\mathbf{x^t}|$ denotes the number of $\mathbf{x^t}$ of the task of $\mathbf{t}$. Eq. 12 leverages the volume-preservation from GIN (Sorrenson et al., 2020).

Similarly, the learning objective for $p(\hat{\mathbf{x}}^\mathbf{t})$ of Eq. 10 from the true observations $\mathbf{x^t}$ also employs MLE:

$$\mathcal{L} = \frac{1}{T} \sum_{t=1}^{T} \Big( \frac{1}{|\mathbf{x^t}|} \sum_{\mathbf{x^t}} \big( \log p(\hat{g}_{-1}(\mathbf{x^t})) \big) \Big) \tag{13}$$

In this equation, following Sorrenson et al. (2020), the volume-preservation is used. When comparing with Eq. 12, we observe that Eq. 13 indicates $\hat{g}$ being trained across all $T$ tasks, highlighting the fundamental distinction between our PTA and ATA approaches.

In addition to Eq. 12 and Eq. 13, our work focuses on identifying and learning the sharing of latent variable $\mathbf{z^t}$. Based on our reinterpretation of Eq. 3, maximizing the distribution similarity of $\mathbf{z^t}$ directly contributes to minimizing the catastrophic forgetting term $\mathcal{F}$. To achieve this objective, for each task $\mathbf{t}$, we employ the KL divergence to further tune both $\hat{g}^{:\mathbf{t}}$ and $g$:

$$\mathcal{KL} = \frac{1}{|\mathbf{x^t}|} \sum_{\mathbf{x^t}} q(\hat{\tilde{\mathbf{z}}}^\mathbf{t}) \log \Big( \frac{q(\hat{\tilde{\mathbf{z}}}^\mathbf{t})}{q(\hat{\overline{\mathbf{z}}}^\mathbf{t})} \Big) = \frac{1}{\mathbf{t}} \sum_{i=1}^{\mathbf{t}} \Big( \frac{1}{|\mathbf{x^t}|} \sum_{\mathbf{x^t}} q(\hat{g}_{-1}(\mathbf{x^t})) \log \Big( \frac{q(\hat{g}_{-1}(\mathbf{x^t}))}{q(\hat{g}_{-1}^{:\mathbf{t}}(\mathbf{x^t}))} \Big) \Big) \tag{14}$$

where $q(\hat{\tilde{\mathbf{z}}}^\mathbf{t})$ and $q(\hat{\overline{\mathbf{z}}}^\mathbf{t})$ denote the posterior of $\hat{\tilde{\mathbf{z}}}^\mathbf{t}$ and $\hat{\overline{\mathbf{z}}}^\mathbf{t}$, which are learned using Eq. 9 and Eq. 11, respectively.

## 5.3 Training and Inference

**Two-stage Training** `ICON` takes inspiration from Li et al. (2024) to implement a two-stage training mechanism. The first stage focuses on independently optimizing the objectives in Eq. 12 and Eq. 13. In the second stage, we jointly train both $\hat{g}^{:\mathbf{t}}$ and $\hat{g}$ by minimizing the KL divergence defined in

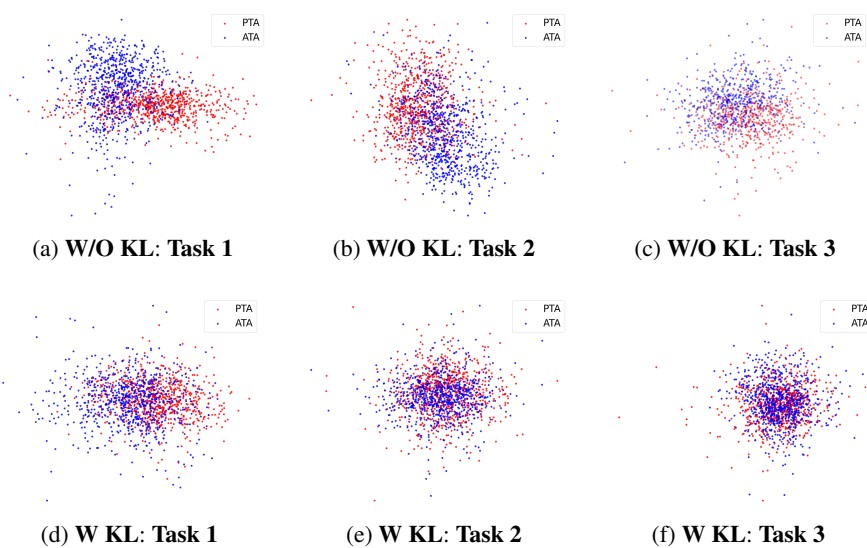

(a) **W/O KL**: **Task 1**          (b) **W/O KL**: **Task 2**          (c) **W/O KL**: **Task 3**

(d) **W KL**: **Task 1**          (e) **W KL**: **Task 2**          (f) **W KL**: **Task 3**

Figure 2: Visualization of latent space distributions across three tasks under PTA (blue) and ATA (red) setups from our simulations. The top row shows representations without optimizing the KL divergence in Eq. 14, displaying significant disparity between PTA and ATA. The bottom row demonstrates improved alignment through our proposed KL-based identification approach, illustrating effective mitigation of catastrophic forgetting across sequential tasks. For clear visualizations, each figure displays 1,000 uniformly sampled points $\hat{\bar{\mathbf{z}}}^{\mathbf{t}}$ and $\hat{\bar{\mathbf{z}}}^{\mathbf{t}}$, respectively.

Eq. 14. This process effectively reduces the discrepancies between the latent representations $\hat{\bar{\mathbf{z}}}^{\mathbf{t}}$ and $\hat{\bar{\mathbf{z}}}^{\mathbf{t}}$. Notably, during training, we process tasks sequentially, maintaining access to the task identity $\mathbf{t}$ to calculate the objective functions in Section 5.2.

**Inference** During inference stage, only $\hat{g}$ is used as the goal of continual learning is to learn a task-invariant mixing function. Thus, **ICON** does not require $\mathbf{t}$ during inference since $\hat{g}$ is task invariant.

## 6 EXPERIMENTS

### 6.1 SYNTHETIC EXPERIMENTS

**Experimental Setup** To evaluate **ICON** 's ability to mitigate catastrophic forgetting ($\mathcal{F}$), we first conducted simulation experiments. We generated synthetic datasets satisfying the identifiability assumptions outlined in Theorem 4.

Specifically, our approach generated four distinct scenarios of observations, with each scenario corresponding to a particular task. The latent variables for each scenario comprised 16 dimensions, which we partitioned into two parts: (1) a 8-dimensional task-invariant component drawn from $\mathcal{N}(0, I)$ that remained constant across all tasks, and (2) a 8-dimensional task-specific component drawn from $\mathbf{z_v} \sim \mathcal{N}(\mu, \sigma^2 I)$, which varied between tasks. For each task, the data generation process begins with $10,000$ latent data points, where $\mu \sim \text{Uniform}(-4, 4)$ and $\sigma^2 \sim \text{Uniform}(0.1, 1)$. Following the practices established in Kong et al. (2022); Li et al. (2023), we used a two-layer MLP to generate the observations, which compriese 16 dimensions.

In our synthetic experiments, we aim to determine whether $\mathcal{F}$ could be minimized using our training objectives presented in Section 5.2. Therefore, we compare the average Root Mean Squared Error (RMSE) of the reconstructions across all 4 tasks obtained from the $\hat{g}$ trained with our objective in Eq. 13, and the mixing function of ATA setup without Eq. 14.

**Results and Discussions** Table 1 summarizes our main findings on our simulations. We evaluate both PTA and ATA setup of **ICON** against the baseline without KL divergence of Eq. 14 in identifying the shared latent varaible $\mathbf{z^t}$.

Table 1: Average Root Mean Squared Error (RMSE) comparison between PTA and ATA frameworks, with and without optimizing KL divergence (Eq. 14). Lower values indicate better performance.

| | | Average RMSE $\times 10^{-1}$ |
|---|---|---|
| PTA setup | w/o KL | 0.12 |
| ATA setup | | 0.20 |
| PTA setup | w KL | 0.12 |
| ATA setup | | 0.13 |

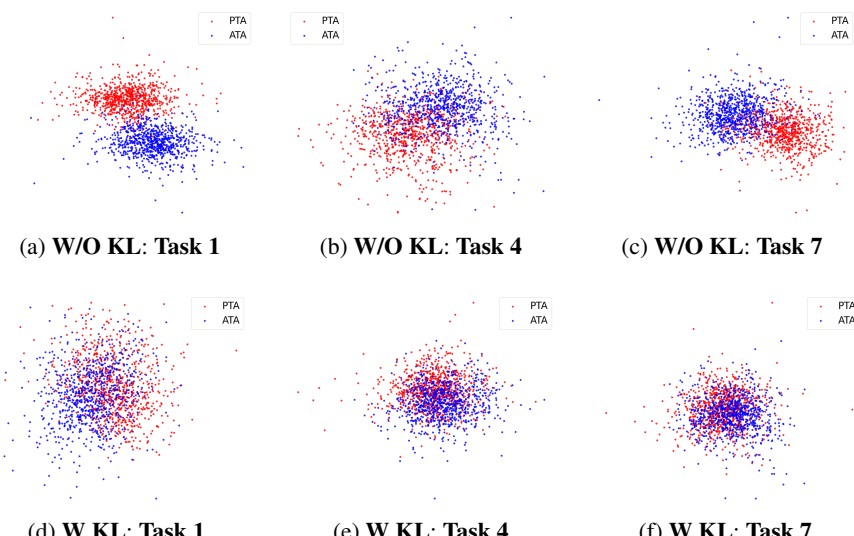

(a) **W/O KL**: Task 1      (b) **W/O KL**: Task 4      (c) **W/O KL**: Task 7

(d) **W KL**: Task 1      (e) **W KL**: Task 4      (f) **W KL**: Task 7

Figure 3: Visualization of latent space distributions across Tasks 1, 4, and 7 on ImageNet-100 dataset, comparing representations from PTA (blue) and ATA (red) frameworks. The top row shows results without using KL divergence optimization in Eq. 14, where significant distribution misalignment indicates catastrophic forgetting as training progresses through tasks. The bottom row demonstrates our **ICON** with KL divergence optimization, exhibiting substantially improved alignment. For visualization clarity, each subfigure displays 1,000 uniformly sampled points from the estimated latent representations $\hat{\bar{\mathbf{z}}}^t$ (PTA framework) and $\hat{\mathbf{z}}^t$ (ATA framework).

We can observe that **ICON** incorporating the KL divergence term substantially improves performance in the ATA setup, reducing the average MSE from 0.20 to 0.13 (a 35% improvement) across all tasks. Notably, this evidently indicates that our approach effectively handles the catastrophic forgetting by identifying the shared latent variables.

Figure 2 provides visual evidence of this improvement compared to the baseline without KL divergence. The latent space distributions across three tasks reveal that in the top row (without KL), there exists significant disparity between the PTA (blue) and ATA (red) representations, particularly pronounced in Tasks 1 and 2. This disparity directly corresponds to catastrophic forgetting in the model. In contrast, the bottom row (with KL) exhibits markedly reduced differences between the PTA and ATA setups. This improved alignment of latent representations confirms that our approach demonstrates its effectiveness in addressing catastrophic forgetting.

## 6.2 REAL-WORLD EXPERIMENTS

To verify the efficacy of our theory in complex real-world scenarios, we further conduct real-work experiments.

**Experimental Setup** We evaluate our approach on two standard benchmarks for handling catastrophic interference: CIFAR-100 (Krizhevsky et al., 2009) and ImageNet-100 (Deng et al., 2009).

Table 2: Average classification accuracy (%) comparison on ImageNet-100 and CIFAR-100 datasets. The best results are highlighted in **bold**, and the second best are in **underline**.

|  | ImageNet-100 | CIFAR-100 |
|---|---|---|
| CoOp (Zhou et al., 2022) | 79.14 | 81.17 |
| MaPLe (Khattak et al., 2023) | 79.23 | 82.74 |
| AttriCLIP (Wang et al., 2023a) | 82.39 | 79.31 |
| Continual-CLIP (Thengane et al., 2022) | 83.99 | 78.65 |
| CLIP-Adapter (Gao et al., 2024) | 84.13 | 78.75 |
| CLAP (Jha et al., 2024) | 87.76 | 86.13 |
| **ICON (ours)** | **88.91** | **87.07** |

CIFAR-100 comprises 60,000 RGB images ($32 \times 32$ pixels) distributed across 100 classes. Following established protocols, we partition this dataset into 10 sequential tasks, each containing 10 distinct classes. Each class contains 500 training and 100 testing samples, ensuring a balanced evaluation framework. For ImageNet-100, a carefully curated subset of the full ImageNet dataset, we utilize higher-resolution images ($224 \times 224$ pixels) from 100 classes. The dataset provides approximately 1,300 training and 50 testing samples per class. Consistent with recent state-of-the-art approach, such as CLAP (Zhou et al., 2022; Thengane et al., 2022; Gao et al., 2024; Wang et al., 2023a; Khattak et al., 2023; Derakhshani et al., 2023; Jha et al., 2024), ImageNet-100 is divided into 10 tasks with 10 classes per task. Across both benchmarks, we report the average classification accuracy on test data across all tasks.

Since both datasets focus on image classification tasks, we adapt **ICON** using noise contrastive estimation (NCE) (Gutmann & Hyvärinen, 2010). First, we train our model using the objectives defined in Equations 12, 13, and 14. Subsequently, we define our NCE loss $\mathcal{L}^c$ as: $\mathcal{L}^c = -\sum_k \log \frac{\exp(\text{sim}(\hat{\mathbf{z}}_k^t, \hat{e}_k)/\tau)}{\sum_m \exp(\text{sim}(\hat{\mathbf{z}}_k^t, \hat{e}_m)/\tau)}$, where $\text{sim}(\cdot, \cdot)$ represents the cosine similarity between the text embeddings $\hat{e}_*$ of the class labels and the learned latent variables $\hat{\mathbf{z}}_k^t$, and $\tau$ is a temperature parameter controlling the sharpness of the distribution. Notably, the text embeddings $\hat{e}_*$ are from all task. Prior to computing $\mathcal{L}^c$, we project $\hat{\mathbf{z}}^t$ to a 512-dimensional embedding space using an MLP layer, aligning with the dimensionality of the text embeddings. For both datasets, we set the dimensionality of the latent representations $\hat{\bar{\mathbf{z}}}^t$ and $\hat{\mathbf{z}}^t$ to 24.

For feature extraction across all tasks and experiments, we employ the Vision Transformer (ViT-B/16) backbone from Radford et al. (2021) to obtain image features $\mathbf{x}^t$, by following Zhou et al. (2022); Thengane et al. (2022); Gao et al. (2024); Wang et al. (2023a); Khattak et al. (2023); Derakhshani et al. (2023); Jha et al. (2024). To ensure fair comparison with prior work, we maintain consistent replay memory configurations. Specifically, we randomly sample 2,000 exemplars for the CIFAR-100 dataset and 1,000 exemplars for the ImageNet-100 dataset. We optimize our network using AdamW Loshchilov & Hutter (2019) with an initial learning rate of 0.002 and weight decay of $10^{-2}$, employing a cosine annealing schedule for learning rate decay. For the contrastive learning objective, we set the temperature parameter $\tau = 0.07$ in the NCE loss $\mathcal{L}^c$. We implement our framework in PyTorch and conduct all experiments on a single NVIDIA GeForce RTX 3090 GPU with 24GB memory.

**Results and Discussions** Table 2 presents our comparative evaluation against state-of-the-art approaches for handling catastrophic interference on ImageNet-100 and CIFAR-100 benchmarks. Our **ICON** demonstrates superior performance across both datasets, achieving 88.91% and 87.07% average accuracy on CIFAR-100 and ImageNet-100, respectively.

On ImageNet-100, **ICON** surpasses the current state-of-the-art method (CLAP) by 1.15%, demonstrating the effectiveness of our identification-based framework on the real-world scenarios. Similarly, **ICON** outperforms the previous best method (CLAP) from 86.13% to 87.07%, representing a substantial margin improvement. The consistent performance improvements across both datasets validate the capability of **ICON** to handle catastrophic forgetting with respect to previous approaches. We trace this capability back to the advantage of identifying shared latent representations between PTA and ATA setup.

Figure 3 provide visual evidence of the effectiveness of **ICON** for catastrophic forgetting handling on the ImageNet-100 dataset. In the top rows (without KL divergence), we observe substantial mis-

Table 3: Average classification accuracy (%) of our ablation studies.

|  | ImageNet-100 | CIFAR-100 |
|---|---|---|
| w/o KL | 74.85 | 75.94 |
| **ICON (ours)** | 88.91 | 87.07 |

(a) **W/O KL**: Task 1     (b) **W/O KL**: Task 4     (c) **W/O KL**: Task 7

(d) **W KL**: Task 1     (e) **W KL**: Task 4     (f) **W KL**: Task 7

Figure 4: Visualization of latent space distributions across Tasks 1, 4, and 7 on CIFAR-100 benchmark, comparing representations from partial-task aware (PTA, blue) and all-task aware (ATA, red) frameworks. The top row displays results without KL divergence optimization (Eq.14), revealing significant distributional misalignment that indicates catastrophic forgetting. The bottom row demonstrates our **ICON** approach with KL divergence optimization, exhibiting substantially improved alignment between PTA and ATA representations. Each subfigure displays 1,000 uniformly sampled points from the estimated latent representations $\hat{\bar{\mathbf{z}}}^t$ (PTA framework) and $\hat{\mathbf{z}}^t$ (ATA framework) for visualization clarity.

alignments between the PTA (blue) and ATA (red) representations. These misalignments indicates the existence of catastrophic forgetting as the distributions occupy distinctly separate regions of the latent space. In contrast, the bottom rows (with KL divergence optimization) exhibits remarkably improved alignments between the PTA and ATA representations across all three tasks. These alignments confirms the superior classification performance, as quantified in Table 2.

**Ablation Studies** In this section, we conduct ablation studies to asses the contribution of KL divergence for solving catastrophic forgetting on the ImageNet-100 and CIFAR-100 datasets.

We summarize the results of our ablation study in Table 3. The results demonstrate that our full ICON framework significantly outperforms the variant without KL divergence optimization across both benchmarks. The dramatic performance gap (88.91% versus 74.85% on ImageNet-100, and 87.07% against 75.94% on CIFAR-100) highlights the critical importance of our KL divergence optimization component, which explicitly maximize the shared latent variables $\mathbf{z^t}$ to the end of minimizing the catastrophic forgetting $\mathcal{F}$ in Eq. 3.

# 7 CONCLUSION

In this paper, we have presented a theoretical framework that characterizes catastrophic forgetting through the lens of identification theory. Upon our identifiability results, we establish a principled approach to mitigating the catastrophic forgetting challenge in continual learning. The empirical results on **ICON** validate our theoretical framework, demonstrating superior performance on both synthetic data and standard continual learning benchmarks.

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

## A  Technical Appendices and Supplementary Material

### A.1  Proof of Theorem 4.1

**Proof:** We proceed our proof by contradiction, focusing on the latent space $\tilde{\mathbf{z}}^{\mathbf{t}}$ in the ATA setting. Suppose that $\mathbf{z}^{\mathbf{t}} \in \mathcal{Z}^{\mathbf{t}}$ simultaneously resides on two distinct manifolds $\tilde{\mathcal{Z}}_1^{\mathbf{t}}$ and $\tilde{\mathcal{Z}}_2^{\mathbf{t}}$. Under this assumption, there exist points $\tilde{\mathbf{z}}_1^{\mathbf{t}} \in \tilde{\mathcal{Z}}_1^{\mathbf{t}}$ and $\tilde{\mathbf{z}}_2^{\mathbf{t}} \in \tilde{\mathcal{Z}}_2^{\mathbf{t}}$ such that we can establish the following:

$$g(\tilde{\mathbf{z}}_1^{\mathbf{t}}) - g(\mathbf{z}^{\mathbf{t}}) = \left( \int_0^1 J_g(\lambda \mathbf{z}^{\mathbf{t}} + (1-\lambda)\tilde{\mathbf{z}}_1^{\mathbf{t}})d\lambda \right) h_1 \tag{15}$$

$$g(\tilde{\mathbf{z}}_2^{\mathbf{t}}) - g(\mathbf{z}^{\mathbf{t}}) = \left( \int_0^1 J_g(\lambda \mathbf{z}^{\mathbf{t}} + (1-\lambda)\tilde{\mathbf{z}}_2^{\mathbf{t}})d\lambda \right) h_2 \tag{16}$$

where $h_1 = \tilde{\mathbf{z}}_1^{\mathbf{t}} - \mathbf{z}^{\mathbf{t}}$, $h_2 = \tilde{\mathbf{z}}_2^{\mathbf{t}} - \mathbf{z}^{\mathbf{t}}$. The L.H.S of Eq.15 and 16 uses the fact that the shared observation can be mapped through either $g$ or $g^{\mathbf{t}}$ from $\mathbf{z}^{\mathbf{t}}$.

We take the substraction of Eq.15 and Eq.16:

$$g(\tilde{\mathbf{z}}_1^{\mathbf{t}}) - g(\tilde{\mathbf{z}}_2^{\mathbf{t}}) = \left( \int_0^1 J_g(\lambda \mathbf{z}^{\mathbf{t}} + (1-\lambda)\tilde{\mathbf{z}}_1^{\mathbf{t}})d\lambda \right) h_1$$
$$- \left( \int_0^1 J_g(\lambda \mathbf{z}^{\mathbf{t}} + (1-\lambda)\tilde{\mathbf{z}}_2^{\mathbf{t}})d\lambda \right) h_2 \tag{17}$$

Let us denote $\Lambda_1 = \left( \int_0^1 J_g(\lambda \mathbf{z}^{\mathbf{t}} + (1-\lambda)\tilde{\mathbf{z}}_1^{\mathbf{t}})d\lambda \right) h_1$, and $\Lambda_2 = \left( \int_0^1 J_g(\lambda \mathbf{z}^{\mathbf{t}} + (1-\lambda)\tilde{\mathbf{z}}_2^{\mathbf{t}})d\lambda \right) h_2$. Eq.17 implies that:

$$||\Lambda_2 - \Lambda_1|| \geq \mathcal{D}(\tilde{\mathcal{Z}}_1^{\mathbf{t}}, \tilde{\mathcal{Z}}_2^{\mathbf{t}})$$
$$\implies ||\Lambda_2|| + ||\Lambda_1|| \geq \mathcal{D}(\tilde{\mathcal{Z}}_1^{\mathbf{t}}, \tilde{\mathcal{Z}}_2^{\mathbf{t}})$$
$$\implies J_{\tilde{\mathbf{u}}}(||h_2|| + ||h_1||) \geq \mathcal{D}(\tilde{\mathcal{Z}}_1^{\mathbf{t}}, \tilde{\mathcal{Z}}_2^{\mathbf{t}})$$
$$\implies \max(||h_2||, ||h_1||) \geq \frac{\mathcal{D}(\tilde{\mathcal{Z}}_1^{\mathbf{t}}, \tilde{\mathcal{Z}}_2^{\mathbf{t}})}{2J_{\tilde{\mathbf{u}}}} \tag{18}$$

where $J_{\tilde{\mathbf{u}}}$ denotes the spectral norm of the Jacobian matrices $\left( J_g(\tilde{\mathbf{z}}_1^{\mathbf{t}}), J_g(\tilde{\mathbf{z}}_2^{\mathbf{t}}) \right)$. This directly contradicts our assumption that $\mathcal{D}(\tilde{\mathbf{z}}^{\mathbf{t}}, \mathbf{z}^{\mathbf{t}}) \leq \frac{\mathcal{D}(\tilde{\mathcal{Z}}_1^{\mathbf{t}}, \tilde{\mathcal{Z}}_2^{\mathbf{t}})}{2J_{\tilde{\mathbf{u}}}}$. Therefore, $\mathbf{z}^{\mathbf{t}}$ can only be explained by a single manifold within $\tilde{\mathcal{Z}}^{\mathbf{t}}$.

Given the injectiveness of $g$ in Eq.2, the correct estimate $\hat{\tilde{\mathbf{z}}}^{\mathbf{t}}$ is feasible for $\tilde{\mathbf{z}}^{\mathbf{t}}$ as Eq.7 suggests. Utilizing an analogous constraint to Assumption 5, we can bound the distance between the estimated latent variables as follows: $\mathcal{D}(\hat{\tilde{\mathbf{z}}}^{\mathbf{t}}, \hat{\mathbf{z}}^{\mathbf{t}}) \leq \frac{\mathcal{D}(\hat{\mathcal{Z}}_1^{\mathbf{t}}, \hat{\mathcal{Z}}_2^{\mathbf{t}})}{2J_{\hat{\mathbf{u}}}}$. $J_{\hat{\mathbf{u}}}$ denotes the spectrum norm of $\left( J_{\hat{g}}(\hat{\mathbf{z}}_1^{\mathbf{t}}), J_g(\hat{\mathbf{z}}_2^{\mathbf{t}}) \right)$.

Extending our contradiction statement, let us denote the difference $\hat{\tilde{\mathbf{z}}}_1^{\mathbf{t}} - \hat{\mathbf{z}}^{\mathbf{t}}$ as $\hat{h}$. Any incorrect estimate $\hat{\tilde{\mathbf{z}}}^{\mathbf{t}}$ would lead to:

$$||\hat{h}|| \geq \frac{\mathcal{D}(\hat{\mathcal{Z}}_1^{\mathbf{t}}, \hat{\mathcal{Z}}_2^{\mathbf{t}})}{2J_{\hat{\mathbf{u}}}} \tag{19}$$

This directly contradicts our established constraint assumption. Therefore, such incorrect estimates are excluded from the feasible solution space.

## B  Implementation Details

We detail the network architectures for both synthetic and real-world experiments in Section 6.1 and 6.2, respectively.

For both synthetic and real-world experiments, we optimize our network using AdamW (Loshchilov & Hutter, 2019) with an initial learning rate of 0.002 and weight decay of $10^{-2}$, employing a cosine annealing schedule for learning rate decay. For the contrastive learning objective, we set the temperature parameter $\tau = 0.07$ in the NCE loss $\mathcal{L}^{\mathbf{c}}$. We implement our framework in PyTorch and conduct all experiments on a single NVIDIA GeForce RTX 3090 GPU with 24GB memory.

## C    THE USE OF LARGE LANGUAGE MODELS (LLMs)

We use LLMs to detect and correct grammatical errors throughout the manuscript.

