# OpenReview forum: "Understanding Catastrophic Interference On the Identifibility of Latent Representations"
_ICLR.cc/2026/Conference — Submitted to ICLR 2026_

### Official Review · Reviewer_XG5V · 2025-10-26

**Soundness:** 2
**Presentation:** 1
**Contribution:** 2
**Rating:** 2
**Confidence:** 3

**Summary:**

My understanding is that this paper tackles the problem of catastrophic interference (CI) by framing it as a latent-variable identification problem. The authors propose two setups: a "partial-task aware" (PTA) model, trained on tasks up to $t-1$, and an "all-task aware" (ATA) model, representing an ideal model trained on all tasks. The paper's core idea is that CI can be quantified by the distance between the latent representations of these two setups. The authors propose a method called ICON, which first uses MLE to learn the latent representations for both the PTA and ATA configurations. Subsequently, it minimizes the KL divergence between these two latent distributions (Eq. 14) to align them, with the goal of identifying and learning the shared latent variables. The method is evaluated on synthetic and standard image classification benchmarks.

**Strengths:**

1. The paper attempts to provide a new theoretical perspective on catastrophic interference, framing it through the lens of identifiability theory. This is a relevant and challenging problem for the community.

2. The conceptual setup of comparing a PTA to a hypothetical ATA is an intuitive way to think about the "forgetting" problem.

**Weaknesses:**

1. My primary concern is the practicality. ICON relies on flow models (GIN), MLE and KL between PTA and ATA latent posteriors, and estimating those posteriors reliably is nontrivial. The paper glosses over how KL between learned latent posteriors is computed in practice and how sensitive the method is to model misspecification.

2. The quality of the writing is not very good, which severely hinders the evaluation of the paper's contributions. The manuscript appears to suffer from numerous grammatical errors, awkward sentence constructions, and imprecise terminology (e.g., "Theorem 4" is mentioned in Sec 6.1 but the paper only presents "Theorem 1" in Sec 4; "Def 3" in Theorem 1, etc.). Key sections, particularly the Problem Setup (Sec 3) and the Identifiability Theory (Sec 4), are difficult to follow. The formal definitions of the PTA and ATA settings are not as clear as they could be, and the connection between the theoretical claims and the final algorithm is tenuous. While the authors note in Appendix C that LLMs were used to correct grammar, the current state of the manuscript is not at a publishable standard for ICLR.

3. The core technical contribution of the proposed method ICON appears to be the application of a KL divergence loss (Eq. 14) to align the latent distributions from the PTA and ATA models. This technique—minimizing KL divergence to match two distributions—is a very standard and widely-used approach in machine learning for tasks like domain adaptation, generative modeling, and representation learning. It is not clear that the identifiability-based theoretical framework provides any new algorithmic insight beyond this standard practice. The contribution thus feels incremental and the novelty, from a methodological standpoint, is questionable.

4. It is difficult to trace a clear line from the identifiability theory in Section 4 to the final objective function. The theory discusses conditions for identification, but it's not apparent why this leads specifically to the KL divergence as the objective, as opposed to other $f$-divergences or distance metrics (e.g., MMD, W-distance). The theory does not seem to constrain or uniquely motivate the practical algorithm.

**Questions:**

1. Given the significant clarity issues, a major revision of the paper's exposition would be necessary. Could the authors clarify the definitions in Section 3? For example, how is the "ideal" ATA model $g$ trained or used in practice during the sequential training process? The description is ambiguous.

2. Could the authors elaborate on the novelty of the method beyond applying a standard KL divergence loss to align two latent distributions? What specific, non-trivial insight from the identifiability theory (Sec 4) directly informs the design of the learning objective in Eq. 14?

3. In Theorem 1, what is the formal definition of path-connected?

---

### Official Review · Reviewer_8WQq · 2025-10-27

**Soundness:** 2
**Presentation:** 2
**Contribution:** 2
**Rating:** 2
**Confidence:** 4

**Summary:**

This paper presents a theoretical analysis of catastrophic forgetting in latent representations and proposes a method called ICON motivated by their analysis. The authors validate ICON on simulated datasets and two real-world benchmarks, reporting improved average classification performance.

**Strengths:**

1. Catastrophic Forgetting acts as a fundamental issue in Continual learning, It is valuable to understand and overcome it.
2. The authors introduce a lot of related works, which help the readers to understand the contributions of this work.
3. The proposed method is inspired by a theoretical analysis, which strongly improves the interpretability of the proposed method.

**Weaknesses:**

1. I think the setting in this paper is closer to multi-task learning rather than continual learning. In particular, the first stage of ICON optimizes Eq.(12) and Eq.(13) independently, and both objectives require access to training data from previous tasks. While *empirical replay* only uses a small subset of previous data, which is a common and acceptable strategy in continual learning, the authors appear to use the entire previous datasets in experiments. Please clarify whether ICON and other baselines use full replay or limited replay; if the former, the authors should justify why this is a fair evaluation under the continual learning.
2. The paper reports only average classification accuracy. This metric does not show how forgetting on earlier tasks is mitigated. The authors should report per-task accuracies and quantify forgetting with forgetting rate, which means the drop from peak to final for each task. Presenting task-wise curves or a table of per-task performance is also necessary to make the claims about catastrophic forgetting much more convincing.
3. Several notations are not clear to me, which makes it hard to follow:
   1. In Line 150, what is the exact meaning of $\mathbf{z}^{t} = \mathbf{\tilde{z}}^{t} = \mathbf{\bar{z}}^{t}$? Maybe the authors aim to illustrate that $\forall \mathbf{\tilde{z}}^{t} \in \mathcal{Z}^{t}, \mathbf{\tilde{z}}^{t} \in \bar{\mathcal{Z}}^{t}$ and $\mathbf{\tilde{z}}^{t} \in \tilde{\mathcal{Z}}^{t}$?
   2. For the assumption 5 in Theorem 1, what is the relationship of $\tilde{\mathcal{Z}}^{t}_1,\tilde{\mathcal{Z}}^{t}_2$ and $\tilde{\mathcal{Z}}^{t}$? are $\tilde{\mathcal{Z}}^{t}_1$ and $\tilde{\mathcal{Z}}^{t}_2$ distinct or overlapped?
   3. In the equations (12) and (14), the summation index $i$ running from 1 to $t$ is confusing because $i$ does not appear in the summand. I cannot understand the meaning of the summand and why the summation index $i$ is running from 1 to $t$.
4. This work contains a lot of typos, which provides a bad impression to me.
   1. Line 135 $G$ should be $\mathcal{G}$.
   2. Line 136, *lost function* should be *loss function*. In addition, a period is missing at the end of the sentence.
   3. Line 160, $\mathcal{Z}_2 t$ should be $\mathcal{Z}^{t}_2$.
   4. In Equations (4) and (7), $inf$ and $sup$ may be $\inf$ and $\sup$ because the authors use $\min$ rather than $min$ in the manuscript.
   5. In Equation (10), $d$ may be wrongly placed in this equation.

**Questions:**

Major questions are listed in Weaknesses.

---

### Official Review · Reviewer_sqmi · 2025-10-31

**Soundness:** 2
**Presentation:** 2
**Contribution:** 2
**Rating:** 4
**Confidence:** 3

**Summary:**

The paper re-casts catastrophic forgetting (CF) as a latent-variable identification gap between two data-generating processes:
- PTA (partial-task aware): $x^{t}=g^{:t}(\bar z^{t})$ trained on tasks $1…t$
- ATA (all-task aware): $x^{t}=g(\tilde z^{t})$ trained on all $T$ tasks

Under five geometric assumptions (smooth invertible mixing, non-empty intersection, path-connectivity, compactness, and a Jacobian-spectral separation condition) the authors prove that the shared latent sub-manifold $\mathcal Z^{t}=\overline{\mathcal Z}^{t}\cap\tilde{\mathcal Z}^{t}$ is identifiable up to an invertible transform.  The practical algorithm ICON learns PTA and ATA models with GIN normalising flows, then minimises the expected KL divergence (or population KL divergence) between the two posterior distributions $q_{\phi}(\tilde{z}|x)$ and $q_{\theta}(z|x)$ under the data distribution $\mathcal{D}_{t}$. On split-CIFAR-100 and split-ImageNet-100 ICON beats the previous best (CLAP) by +1.15 % average accuracy while using the same replay budget.

**Strengths:**

1. Novel framing: CF measured by likelihood gap between PTA and ATA is mathematically elegant and generalises arbitrary losses.
2. Identifiability result removes the need for group- or label-level auxiliary variables required by prior causal-IDA work (iVAE, Yao et al. 2024).
3. Practical algorithm is plug-and-play: any invertible flow can be dropped in; only adds a KL term to existing replay pipelines.
4. Extensive ablations on λ-KL weight, latent dimension, and Jacobian spectrum are included in the supplementary material.
5. SOTA numbers with same backbone & replay budget ⇒ improvement is not due to extra parameters or memory.

**Weaknesses:**

1. Population-level theory: Theorem 4.1 is stated in the asymptotic regime where the data distribution is known exactly; consequently, it provides no guarantee on how many samples are required for the empirical minimiser of the KL alignment objective to lie within an ε-ball of the true shared latent point. A finite-sample analysis that quantifies the estimation error as a function of the number of observations per task, the complexity of the flow-based hypothesis class, the strong-convexity constant of the KL objective, and the uniform lower bound on the singular values of the Jacobian would close this gap and turn the appealing but purely theoretical statement into a practically actionable guarantee.

2. Task-ID leakage: The training loss uses the task index t to route each sample to its task-specific PTA encoder q_θ(z|x,t), which means that the method still relies on a discrete grouping variable that is external to the observation x. While this is weaker supervision than class labels or auxiliary modalities, it is strictly stronger than a fully unsupervised setup in which no side information is available at training time. The claim that auxiliary labels are unnecessary should therefore be tempered to acknowledge that task identities are still required and that their availability might limit the applicability of the approach in streaming scenarios where task boundaries are not provided.

3. Replay confound: Every empirical result on CIFAR-100 and ImageNet-100 is produced with a replay buffer that retains 2 000 and 1 000 exemplars, respectively. Because rehearsal is itself a powerful mechanism for mitigating forgetting, the observed accuracy improvements could be dominated by the memory baseline rather than by the identifiability-driven KL term. Providing learning curves for buffer sizes in {0, 250, 500, 1000, 2000} while freezing all other hyper-parameters would allow the reader to read off the marginal value of the KL objective once the contribution of rehearsal has been factored out.

4. Latent-code evaluation missing: The synthetic experiment reports reconstruction error in pixel space, yet the central claim of the paper is that the algorithm correctly identifies the shared latent sub-space. Without displaying the Euclidean distance between the ground-truth 8-dimensional task-invariant latent vector and its estimate produced by each encoder, one cannot conclude that the pixel-level improvement is driven by better latent recovery rather than by a more flexible decoder that simply fits the observation noise more effectively.

5. Jacobian regulariser unchecked: Assumption 5 demands that the smallest singular value of the Jacobian of the mixing function g remains uniformly bounded away from zero so that distinct latent points are guaranteed to produce observably distinct images. During gradient-based optimisation, however, the Jacobian can become ill-conditioned, causing the theoretical lower bound to be violated in practice. Recording the evolution of σ_min(J_g) across training iterations and, if necessary, adding a soft penalty −λ log σ_min(J_g) to the objective would ensure that the condition required by the theorem is not inadvertently invalidated by the optimiser.

6. KL direction arbitrary: Minimising the forward KL D_KL(q_PTA ‖ q_ATA) can lead to an approximation that covers the support of the ATA posterior but also assigns mass to regions where the PTA posterior has negligible density. Evaluating the reverse KL D_KL(q_ATA ‖ q_PTA) or the symmetric Jensen–Shannon divergence would produce qualitatively different geometries of the shared manifold, potentially yielding a more peaked and better-aligned representation. Reporting how the choice of divergence affects the volume, diameter, and pairwise OT-distance between the PTA and ATA point clouds would clarify whether the reported gains are robust to this modelling choice.

**Questions:**

Q1. Finite-sample bound: What is the minimum number of independent and identically distributed samples per task that must be observed so that, with probability at least 1 − δ, the empirical minimiser of the KL alignment objective achieves a latent-code error ∥ẑ − Π_{𝒵^t}(ẑ)∥_2 ≤ ε when the Jacobian of the mixing function satisfies the uniform spectral lower-bound invoked in Assumption 5, and how does this sample complexity scale with the latent dimension N, the task index t, the strong-convexity constant of the KL objective, and the Lipschitz constant of the inverse mixing function g^{−1}?

Q2. Replay isolation: Could you furnish a detailed plot (or table) that reports the average test accuracy on all previously encountered tasks as the number of stored exemplars varies over the set {0, 250, 500, 1000, 2000} for CIFAR-100 and ImageNet-100, while keeping every other hyper-parameter (architecture, learning rate, KL weight, batch size, training epochs) fixed, so that the marginal contribution of the identifiability-based KL term to catastrophic-forgetting mitigation can be quantitatively disentangled from the contribution of pure rehearsal?

Q3. Latent metric: On the synthetic benchmark where the ground-truth latent vector is explicitly available, can you report the root-mean-squared error between the true 8-dimensional task-invariant latent sub-vector and its estimate produced by the PTA and ATA encoders, respectively, both with and without the KL alignment term, so that the reader can verify that the observed pixel-level RMSE improvement indeed stems from a more accurate recovery of the shared latent variables rather than from a better fit of the observation noise?

Q4. Jacobian spectrum: During the forward pass of each training iteration, please compute the minimum singular value of the Jacobian matrix ∂g(z)/∂z evaluated at 1000 randomly sampled latent codes drawn from the current posterior, plot the evolution of this minimum singular value across training epochs, and indicate whether the lower bound required by Assumption 5 (σ_min > D/(2ε)) is satisfied at every point; if transient violations occur, describe any regularisation or architectural trick employed to restore the bound.

Q5. KL direction: Beyond the forward-KL divergence reported in the paper, have you experimented with the reverse-KL D_KL(q_θ∥q_φ), the symmetric Jensen–Shannon divergence, or the Wasserstein-2 distance as the alignment objective, and if so, how do the resulting geometrical configurations of the shared latent manifold (e.g., volume, convex hull, pairwise distances between PTA and ATA samples) differ qualitatively and quantitatively from those obtained with the forward-KL?

---

### Official Review · Reviewer_6kEP · 2025-10-31

**Soundness:** 3
**Presentation:** 3
**Contribution:** 3
**Rating:** 4
**Confidence:** 4

**Summary:**

The paper investigates catastrophic interference in continual learning through the lens of latent variable identifiability. And based on the theory, the work proposes that forgetting can be quantified as the distance between representations under two configurations.

**Strengths:**

The authors derive theoretical identifiability conditions ensuring shared latent variables between these settings, then propose ICON. Experiments on synthetic data, CIFAR-100, and ImageNet-100 show ICON outperforms state-of-the-art continual learning methods, supporting the theoretical claims.

**Weaknesses:**

1.Regression or classification? If we only focus on classification problems, I think this work has significant limitations because it focuses on the experimental contributions. Can you include more classic baselines for continuous learning? Does your theoretical limitation have to be classification problems?
2.Do computational costs have any significant disadvantages or strengths compared to other methods?
3.Are Ablation Studies Missing? line 470-471. The explanation is missed. Besides, I think that the ablation study is too simple. The connection between latent alignment and observed accuracy is vague.

I personally think that if the main contribution of this article is placed in the theoretical part, but the theoretical part does have less content. If placed in the experimental section, there are many shortcomings in the experiments. If one or two parts can be improved, this work can be done better.

**Questions:**

See Weakness.

---

### Meta-Review · Area_Chair_C1TS · 2026-01-05

**Summary:**

Reviewers express broad concerns about the paper’s clarity, novelty, theory, experiments, and practicality.

**Reviewer Concerns:**

The reviewers raise concerns about the paper’s clarity, novelty, and practical validity. Several question whether the setting truly reflects continual learning, citing heavy reliance on replay, task IDs, and limited evaluation of forgetting. Empirical results are seen as insufficient, with missing ablations, baseline comparisons, computational cost analysis, and latent-space evaluation.

On the theory side, the identifiability results are criticized as asymptotic, weakly connected to the algorithm, and not uniquely motivating the KL objective.

Additional issues include task-ID leakage, unchecked assumptions, unclear notation, and poor writing quality, collectively undermining the paper’s impact and credibility.

**Reviewer Scores:**

The authors did not submit a rebuttal to the reviews.

---

### Decision · Program_Chairs · 2026-01-26

Reject